# Fetal *RHD* Screening in RH1 Negative Pregnant Women: Experience in Switzerland

**DOI:** 10.3390/biomedicines11102646

**Published:** 2023-09-27

**Authors:** Bernd Schimanski, Rahel Kräuchi, Jolanda Stettler, Sofia Lejon Crottet, Christoph Niederhauser, Frederik Banch Clausen, Stefano Fontana, Markus Hodel, Sofia Amylidi-Mohr, Luigi Raio, Claire Abbal, Christine Henny

**Affiliations:** 1Interregional Blood Transfusion SRC Berne Ltd., 3008 Berne, Switzerland; 2Institute for Infectious Diseases, University of Berne,3010 Berne, Switzerland; 3Department of Clinical Immunology, Copenhagen University Hospital—Rigshospitalet, 2100 Copenhagen, Denmark; 4Faculty of Biology and Medicine, University of Lausanne, 1005 Lausanne, Switzerland; 5Department of Obstetrics and Gynecology, Cantonal Hospital Lucerne, 6000 Lucerne, Switzerland; 6Department of Obstetrics and Gynecology, University Hospital of Berne—Inselspital, 3010 Berne, Switzerland; 7Division of Hematology, Lausanne University Hospital—CHUV, 1011 Lausanne, Switzerland

**Keywords:** RH1 incompatibility, hemolytic disease of the fetus and newborn (HDFN), noninvasive prenatal diagnosis, cell-free fetal DNA, RH immunoglobulin prophylaxis

## Abstract

RH1 incompatibility between mother and fetus can cause hemolytic disease of the fetus and newborn. In Switzerland, fetal *RHD* genotyping from maternal blood has been recommended from gestational age 18 onwards since the year 2020. This facilitates tailored administration of RH immunoglobulin (RHIG) only to RH1 negative women carrying a RH1 positive fetus. Data from 30 months of noninvasive fetal *RHD* screening is presented. Cell-free DNA was extracted from 7192 plasma samples using a commercial kit, followed by an in-house qPCR to detect *RHD* exons 5 and 7, in addition to an amplification control. Valid results were obtained from 7072 samples, with 4515 (64%) fetuses typed *RHD* positive and 2556 (36%) fetuses being *RHD* negative. A total of 120 samples led to inconclusive results due to the presence of maternal or fetal *RHD* variants (46%), followed by women being serologically RH1 positive (37%), and technical issues (17%). One sample was typed false positive, possibly due to contamination. No false negative results were observed. We show that unnecessary administration of RHIG can be avoided for more than one third of RH1 negative pregnant women in Switzerland. This reduces the risks of exposure to a blood-derived product and conserves this limited resource to women in actual need.

## 1. Introduction

Following ABO, the RH blood group system is the second most important blood group system in transfusion medicine due to its high potential for alloimmunization. The RH1 (also referred to as RhD) antigen, encoded by the *RHD* gene, is highly immunogenic. In alloimmunized RH1 negative mothers, the anti-RH1 antibody can cause hemolytic disease of the fetus and newborn (HDFN) [1]. Alloimmunization of RH1 negative women during pregnancy can be prevented by the administration of RH immunoglobulin (RHIG). Postnatal anti-D prophylaxis for RH1 negative mothers with RH1 positive newborns was introduced in the late 1960s [2,3] and reduced the risk of alloimmunization from approximately 15% to less than 2% [4]. In the 1990s, many countries introduced routine antenatal anti-D prophylaxis (RAADP) for RH1 negative pregnant women. Consequently, the combination of antenatal prophylaxis in the third trimester of the pregnancy and postnatal administration further reduced the risk to less than 0.5% [5,6,7,8].

In the Caucasian population, around 15% of all mothers are RH1 negative. Of these, around 38% carry a RH1 negative fetus and thus will receive RHIG through RAAPD, although they are not at risk of alloimmunization [9]. RHIG is a blood product and its use has two major drawbacks. First, its application exposes RH1 negative women to unnecessary risks, as seen with hepatitis C infections from contaminated anti-RH1 products [10]. Second, the production of RHIG involves the immunization of RH1 negative plasma donors with red blood cells collected from RH1 positive individuals [11]. This is an ethically questionable practice if the use of RHIG could be optimized by targeted administration without an increased risk for pregnant women.

The discovery of cell-free fetal DNA (cffDNA) in the maternal circulation [12] enabled the development of non-invasive fetal genotyping by real-time PCR-based detection of *RHD* specific sequences [13], providing a safe method to determine the fetal *RHD* status. Sufficient amounts of cffDNA for the amplification of *RHD* sequences are present as early as gestation week 11 and maximum sensitivity is reached in week 24 [14]. In 2010, Denmark introduced the first national screening program intended for targeted antenatal anti-RH1 prophylaxis [15]. Similar initiatives followed in the Netherlands in 2011 [16], in Finland in 2014 [17] and in Norway in 2016 [18]. Regional screenings were established in the UK, in Sweden, France, Belgium and recently in northern Italy [19,20]. In Switzerland, noninvasive prenatal *RHD* screening in gestation weeks 18 to 24 has been recommended since 2020, with the aim of changing from general application of RHIG to all RH1 negative mothers to targeted administration to only those carrying a RH1 positive fetus [21]. The test regime includes a general checkup in early pregnancy between gestation weeks eight and twelve, with serological typing of ABO and RH1 antigens. Women shown to be RH1 negative are then given the possibility of prenatal fetal *RHD* screening. Costs for this diagnostic procedure are covered by the obligatory basic health insurance.

An in-house developed screening method for detection of fetal *RHD* exons 5 and 7 by real-time PCR is used in our institute. Here, we present a detailed analysis of all consecutive samples received between January 2020 and June 2022, corresponding to more than 7000 screens performed in this period of 30 months. Our data show that unnecessary antenatal administration of RHIG prophylaxis could be avoided in more than one third of the pregnancies.

## 2. Materials and Methods

### 2.1. Preparation of Cell-Free DNA from Plasma Samples

Prenatal noninvasive *RHD* screening is not centralized in Switzerland. In our diagnostic routine, we request a blood sample of pregnant women who have been confirmed as serologically RH1 negative. The samples must be collected in tubes containing EDTA or in Cell-Free DNA BCT^®^ tubes (Streck, La Vista, NE, USA). Tubes must be stored and shipped at room temperature. Samples exceeding the limit of 5 days between blood withdrawal and delivery are rejected. Plasma preparation of the first 1356 samples reported here was performed as follows: Plasma was separated from cells by centrifugation at 2500× *g* for 10 min and transferred to a new tube. From there, two aliquots of 650 µL each were taken and centrifuged in individual tubes at 16,000× *g* for 15 min. Subsequently, 600 µL of the supernatant of each tube was collected and combined to 1.2 mL in a single tube for automated DNA extraction (see below). To lower the handling time with increasing sample numbers and to reduce the risks of sample mix-up and contamination, an alternative protocol for plasma separation was validated and applied for the remainder of the reported analyses. Plasma was collected in cryotubes after a single step of centrifugation at 2500× *g* for 10 min. A retain sample of 1.2 mL was taken and stored at −70 °C if plasma amounts were sufficient. The remaining plasma (a minimum of 1.2 mL) was either processed immediately or stored at −70 °C until the analysis. The remaining blood in the collection tube containing erythrocytes, buffy coat and traces of plasma was resuspended and aliquots were stored at 4 °C for serological analyses and at −20 °C for genetic analyses, if necessary. A minimum of 1.2 mL plasma was subjected to automated extraction of cell-free fetal DNA using the QIASymphony DSPVirus/Pathogen Kit (Qiagen, Hilden, Germany) on a QIASymphony SP platform, processing 1 mL of plasma with a final elution volume of 60 µL.

### 2.2. Fetal RHD Genotyping

Determination of the fetal RHD genotype was performed on a ViiA 7 Real-Time PCR System (Applied Biosystems, Foster City, CA, USA). Primers and probes for the detection of RHD exon 5, RHD exon 7 and β-globin exon 1 were published previously [22,23,24]. In brief, exon 5 and 7 were simultaneously detected with the primer/probe combinations RhD exon 5F 5′-CGCCCTCTTCTTGTGGATG-3′, RhD exon 5R 5′-GAACACGGCATTCTTCCTTTC-3′, RhDexon5Probe 5′-VIC-TCTGGCCAAGTTTCAACTCTGCTCTGCT-TAMRA-3′ and Rhesus-D-940S 5′-GGGTGTTGTAACCGAGTGCTG-3′, Rhesus-D-1064AS 5′-CCGGCTCCGACGGTATC-3′, Rhesus-D-968 5′-FAM-CCCACAGCTCCATCATGG GCTACAA-TAMRA-3′, respectively. Amplification and detection of exon 1 of β-globin in total DNA with primers β-globin-F 5′-GTGCACCTGACTCCTGAGGAGA-3′, β-globin-R 5′-CCTTGATACCAACCTGCCCAG-3′ and probe Beta-globin-402-T 5′-FAM- AAGGTGAACGTGGATGAAGTTGGTGG-TAMRA-3′ served as control. All primers and probes were used at a final concentration of 300 nM and 100 nM, respectively. In addition, the reactions contained 10 µL DNA template, 1 x PCR buffer, 4 mM MgCl_2_, 200 µM each dNTP, 1 x ROX passive reference dye and 2 units HotStarTaq DNA Polymerase (Qiagen) in a total volume of 50 µL. The PCR program included an initial denaturation step at 95 °C for 10 min followed by 50 cycles of 15 s at 96 °C and 60 s at 60 °C. Reactions for amplification of RHD exons were run in triplicate. The control targeting β-globin was set up in duplicate reactions. Additional controls included a non-template reaction, as well as positive and negative controls, with cell-free fetal DNA prepared from plasma pools with known RHD genotype according to previous analyses. Individual patient samples were evaluated only if the three different run controls showed the expected and valid results. The following criteria were applied for the evaluation of patient samples: β-globin had to be amplified successfully in both reactions with a cycle threshold (Ct) value lower than 33.5 ensuring sufficient yield of total DNA. A fetus was considered RHD negative if a maximum of one out of the six reactions led to the amplification of an RHD exon with a Ct of ≤ 42. In contrast, fetuses were defined as RHD positive if, in at least five out of six reactions, RHD exons were amplified. All other results of amplification of RHD exons were considered either invalid (two out of six) or initially inconclusive (three or four out of six). Analyses with invalid results had to be repeated. Initially, inconclusive results of RHD exon amplification required extended manual evaluation and could be considered RHD positive if at least three out of six reactions showed amplification according to our internal regulations. Furthermore, amplification of RHD exons with Ct values close to values obtained with β-globin (ΔCt ≤ 2.5) indicated the presence of a maternal RHD variant. Such analyses were classified as inconclusive. All analyses were initially technically validated by the executing staff. In a second step, an in-house generated software tool was used to integrate patient information and computer assisted analysis of results and validation.

### 2.3. Validation

Before routine implementation of the aforementioned protocol, the method was validated with two dedicated sets of plasma samples. Set A consisted of 150 plasma samples collected at the University Hospitals of Berne (Inselspital) and Lausanne (CHUV) in gestation weeks 12 to 40. The cell-free DNA of these samples was extracted manually using the QIAamp DSP Virus Kit (Qiagen) and the fetal *RHD* status was confirmed by postnatal serological analysis from umbilical cord blood. Sample set B was collected at the Department of Clinical Immunology of the Copenhagen University Hospital in Denmark. It consisted of 365 anonymized plasma samples with known fetal *RHD* status as determined by a method similar to the presented. The majority of these samples were collected in gestation week 25. The cell-free DNA was extracted automatically on a QIASymphony SP platform as described above.

## 3. Results

The validation of our method revealed high diagnostic specificity and sensitivity of the test. Specificity was determined by the rate of true negative results and was calculated after analyzing 208 *RHD* negative samples consisting of 54 samples of set A and 154 samples of set B. Out of all 208 samples, one sample of set A was classified false positive, possibly due to the presence of a fetal *RHD* variant. Hence, our test method had a specificity of 99.5%. Sensitivity was defined by the rate of true positive results and was calculated after testing 96 samples of set A and 211 samples of set B as *RHD* positive. All 307 samples were correctly classified, resulting in a sensitivity of 100%. Sample set B was further used to test the reproducibility by comparing the results of our fetal genotyping to the reported results. No discrepancies were observed; therefore, reproducibility was 100%. Additional parameters, including analytical sensitivity, measuring range and linearity, intra-assay precision, inter-assay precision and robustness, were tested at a later time point according to recent recommendations for fetal genotyping [25]. All results were within the range of acceptance, and requirements were met. The general recommendation to analyze the fetal *RHD* status in RH1 negative women as the basis for deciding to administer RHIG prophylaxis was implemented in Switzerland in 2020 [21]. Since then, more than 9000 tests have been performed in our institute. Figure 1 provides an overview of the results of 7192 individual analyses performed between January 2020 and June 2022.

More than 90% of all analyzed samples were collected in gestation week 18 or later, with a median and average gestational age of 22 (Figure 1A). Our institute accepts samples for fetal genotyping from 11 + 1 weeks of gestation on. However, a second test is advised in early pregnancy if fetuses are tested *RHD* negative to confirm the result after gestation week 18 with higher amounts of cffDNA present (see also Discussion).

The vast majority of the 7072 analyses revealed valid results, as shown in Table 1 and Figure 1B. Of these, 2556 fetuses (36.1%) tested *RHD* negative and consequently, administration of RHIG was not indicated. A total of 4515 fetuses were shown to be *RHD* positive and the administration of prophylaxis was recommended. A single analysis led to a false positive result. The fetus was classified as *RHD* positive after detection of exon 5 in triplicate and two positive reactions for exon 7. However, postnatal serological analysis revealed that the newborn was RH1 negative. This discrepancy was most likely due to contamination or a sample mix-up (see Discussion). Neonatal cord blood typing for RH1 is optional, according to the Swiss recommendations. We have no information on how often such postnatal analyses were performed and whether more discrepancies exist. The University Hospital of Berne and the Cantonal Hospital Lucerne provided us with serological data for more than 150 cases included in the presented dataset. The *RHD* status of all fetuses in analyses with valid results was typed correctly. One fetus was undetermined due to the presence of a maternal *RHD* variant and RHIG prophylaxis was recommended. Postpartum serologic analysis showed that the newborn was RH1 negative. Based on the presented data, the overall sensitivity was 100% (95%CI 99.92–100%) and the specificity was 99.96% (95%CI 99.78–100%).

Inconclusive or invalid results were obtained in 120 tests, corresponding to 1.7% of the total amount of performed analyses. Reasons for failed tests were the detection of maternal *RHD* exons in 90 cases, followed by technical problems (n = 21) and the presence of *RHD* gene variants in the fetus (n = 9), as presented in Figure 1B. Detection of maternal signals for *RHD* initiated further serological tests using the stored blood sample obtained during the process of plasma preparation. Here, 44 pregnant women tested RH1 positive, making RHIG prophylaxis unnecessary. Out of the remaining 46 samples, 11 samples were further investigated with the attempt to identify the variant alleles. These results are presented in Table 2, together with cases without further investigation of variant alleles. The presence of the allele *RHD*01W.01* was found in three women. This allele leads to a reduced expression of RH1 with all its immunogenic epitopes. Hence, carriers of this allele are considered RH1 positive, making prophylaxis by RHIG unnecessary in these cases [26,27]. For all other women with detectable maternal *RHD* exons, administration of RHIG was recommended.

The presence of fetal *RHD* variants became evident in nine cases (Figure 1B). As a common result, only the fetal exon 7 was amplified, whereas the fetal exon 5 was not detected. A likely explanation for this phenomenon is the presence of the allele *RHD*08N.01*. This allele carries a 37 bp insertion in intron 3 and several single nucleotide variations (SNVs) in exons 4, 5 and 6, preventing the detection of exon 5 by the primer/probe combination used in our analysis. In one case, genotyping of the father showed the presence of *RHD*08N.01,* suggesting that the fetus inherited this allele; therefore, the fetus was determined as *RHD* negative and prophylaxis was not indicated. In all other cases, samples of the fathers were not provided. Hence, the *RHD* status of the fetuses could not be determined with certainty and prophylaxis was recommended.

Technical problems led to invalid or inconclusive results in 21 cases. Of these, seven analyses could be successfully repeated after a second DNA extraction using the retain sample. Reasons for repetition of these tests were either failed PCR with no amplification at all (n = 4) or tests with ambiguous results detecting two, three or four out of six *RHD* exons (n = 1 each). In three out of the 21 cases, invalid results were obtained due to insufficient amounts of total DNA (n = 1) or amplification of *RHD* exons in either two or three out of six reactions. These three tests were not repeated, since no additional blood sample was received; thus, administration of RHIG was recommended. The remaining eleven tests were classified as invalid for several reasons, among them putative contamination (n = 5), low quality of extracted DNA (n = 4), technical problems in sample processing (n = 1) and one case of mix-up leading to the receipt of a wrong sample. In all of these cases, an additional analysis with new samples collected later in the pregnancy led to valid results.

In summary, the results of the presented analyses revealed that unnecessary RAADP could be avoided for 2556 RH1 negative pregnant women carrying a *RHD* negative fetus. This number is expected, given that about one third of RH1 negative women carry RH1 negative fetuses in the Caucasian population [9]. Of all 7192 tests performed, a small amount of 120 led to inconclusive or invalid results. Deducting the number of 44 women who later tested RH1 positive and thus were not in need for RAADP and considering a proportion of 36.1% of *RHD* negative fetuses as determined in our analysis (Table 1), 28 additional analyses would have revealed a *RHD* negative fetus. Hence, for all 2584 women not needing RHIG prophylaxis, unnecessary treatment could be avoided in 98.9% of the cases.

## 4. Discussion

We have established a system for routine screening of fetal *RHD* genotypes in RH1 negative women according to the Swiss recommendation of 2020 [21] and applied the method successfully in our laboratory in more than 9000 cases to date. This article summarizes the results of 7192 analyses performed in a period of 30 months. The vast majority of 7072 led to valid test results and 120 tests were initially inconclusive or invalid for reasons shown in Figure 1B. Only one single false positive result was reported to us. Our analysis suggested that the fetus was *RHD* positive, whereas postnatal serological analysis showed that the newborn was RH1 negative. The analysis was repeated twice after extraction of DNA from retained frozen plasma samples, resulting in one or two detected amplicons of *RHD* exons. The discrepancy with the initially obtained results is best explained with a pre-analytical contamination. According to the general recommendations in Switzerland, serological testing of RH1 of newborns is not mandatory. There are no statistics regarding how often it is performed; more discrepancies might exist. False positive results lead to unnecessary administration of RHIG. Routine prophylaxis used to be the standard procedure for all RH1 negative women before 2020 and thus is not regarded as a severe outcome.

False negative results, in contrast, exclude RH1 negative mothers from prophylaxis required in pregnancies with RH1 positive fetuses and may lead to morbus hemolyticus neonatorum (MHN) in the case of alloimmunization. No false negative results were reported. False negative results may be a consequence of insufficient amounts of fetal DNA extracted from maternal plasma. The amount of cffDNA in the maternal circulation increases during pregnancy. To ensure high sensitivity, it is recommended to perform genotyping between gestation weeks 18 and 24 in Switzerland. Accordingly, 93.9% of the analyzed samples were collected in gestation week 18 or later (Figure 1A). A total of 421 samples (5.9%) were collected in gestation weeks 11 to 17 and successfully analyzed without invalid results. However, it was recommended to repeat the analyses with *RHD* negative fetuses (n = 144) at a later time point during the pregnancy. Of these, 49 analyses (34%) were repeated using samples collected after gestation week 18 and no discrepancies to the previous results were observed.

The presented test strategy includes a control for extraction and amplification of total DNA by the detection of *β-globin*. Additionally, other housekeeping genes including C-C chemokine receptor type 5 (*CCR5*), glyceraldehyde 3-phosphate dehydrogenase (*GAPDH*), superoxide dismutase (*SOD*) and albumin (*ALB*) are described in the literature as controls, as summarized in [29]. Alternative methods use the amplification of internal controls such as synthetic oligonucleotides [30] or DNA from other species [20] that are added during the extraction. Controlling the yield of fetal DNA is less straightforward. In pregnancies with male fetuses, the sex determining region Y (*SRY*) gene can be amplified. However, this control is only applicable in around 50% of the cases. The Ras association domain family member 1 (*RASSF1*) gene, in contrast, can be detected in both genders. Due to epigenetic regulation, the promoter of this gene is hypermethylated in fetal DNA and hypomethylated in maternal cells, leading to differential sensitivity to restriction digest with the enzyme BstUI [31]. This provides an elegant way to detect exclusively fetal DNA derived signals after treatment of total DNA with BstUI digesting maternal template DNA only. However, in our hands, this procedure proved too unreliable and required too much handling time to be included in our routine analyses with high sample numbers. Thus, the presented method relies only on the amplification of *β-globin*. This is in accordance with findings demonstrating that this procedure is acceptable [32]. Only 4 results out of the 7192 reactions were invalid due to failed amplification of *β-globin*. Failed PCR reactions may result from poor extraction of template DNA due to technical problems during the procedure. Consequently, both human DNA and spiked control DNA would not be amplified. However, failure could also be caused by nuclease contamination during plasma preparation. In that case, addition of an internal control in later steps of the protocol and its subsequent successful amplification in the absence of cell free DNA might lead to false negative results. Therefore, we prefer to use a housekeeping gene as control, offering more criteria to assess the quality of extracted template DNA.

Various protocols are applied for fetal genotyping that mainly differ in the DNA extraction procedure, amplification control and the number of *RHD* exons being detected [29]. A single exon approach detecting *RHD* exon 4 is used successfully in the nationwide screening program in Sweden [33], whereas most other approaches detect *RHD* exons 5 and 7 or a combination of 5, 7 and 10. Combinations of two or more exons are meant to minimize the risk for obtaining false negative results due to the undetected presence of *RHD* variants that ultimately might cause RH1 immunization. By targeting *RHD* exons 5 and 7, fetal variants were detected in nine cases based on the amplification of fetal exon 7 only. A likely explanation for this result is the presence of allele *RHD*08N.01,* which is a major cause of RH1 negativity in people of African descent. Another possibility is the presence of the allele *RHD*06.* Carriers of this hybrid allele possess exons 4 and 5 of the closely related *RHCE* gene. In both scenarios, *RHD* exon 7 can be amplified, whereas exon 5 is excluded due to the existing SNVs. A high number of variant *RHD* alleles contain SNVs in exon 5. Without further analysis, a specific genotype cannot be defined and prophylaxis is recommended. It is worth mentioning that exon 4 would not be amplified using the single exon approach in the case of the presence of *RHD*06* and the fetus would be typed false negative.

Maternal *RHD* DNA was detected in 90 samples (Figure 1B). In such cases, a serological analysis was performed using the stored erythrocytes. Here, 44 women were shown to be RH1 positive, making further analyses and RHIG prophylaxis unnecessary. Of the remaining 46 samples, 29 were positive for both exons 5 and 7, whereas in 17 analyses only exon 7 could be amplified (Table 2). If *RHD* exons were detected in the maternal DNA, the fetal genotype could not be determined and RHIG prophylaxis was recommended. A subset of 11 samples was further analyzed to identify the maternal variant alleles. Of those, *RHD*01W.01* was the most prevalent, with three women identified as carriers. Although these individuals were regarded as RH1 positive and thus did not require RHIG prophylaxis, the low number of maternal variants identified in general and the frequency of *RHD*01W.01* did not justify extended routine analyses. Detection of maternal variants routinely leads to the recommendation of RHIG prophylaxis. Extended genotyping to define the maternal variant alleles is only performed upon request.

In summary, we established a robust method for highly sensitive and specific screening of fetal *RHD* for targeted RHIG prophylaxis. Based on the obtained results, unnecessary administration could be avoided in more than 98% of those pregnancies where prophylaxis would not have been indicated. This corresponds to more than 2500 pregnancies over the course of 30 months. This number is well in agreement with data reported in the pioneering studies performed in Denmark, the Netherlands and Finland, showing that more than one third of RH1 negative pregnant women do not require anti-RH1 prophylaxis [15,16,17]. RHIG is a blood product prepared from plasma of hyperimmunized donors, thereby raising ethical concerns and bearing a potential risk of infectious diseases for the recipients [10]. The recommended screening program will enable targeted administration of RHIG. This will contribute to reduce the unnecessary use of this limited resource and the exposure to the intrinsic risk of blood-derived products.

## Figures and Tables

**Figure 1 biomedicines-11-02646-f001:**
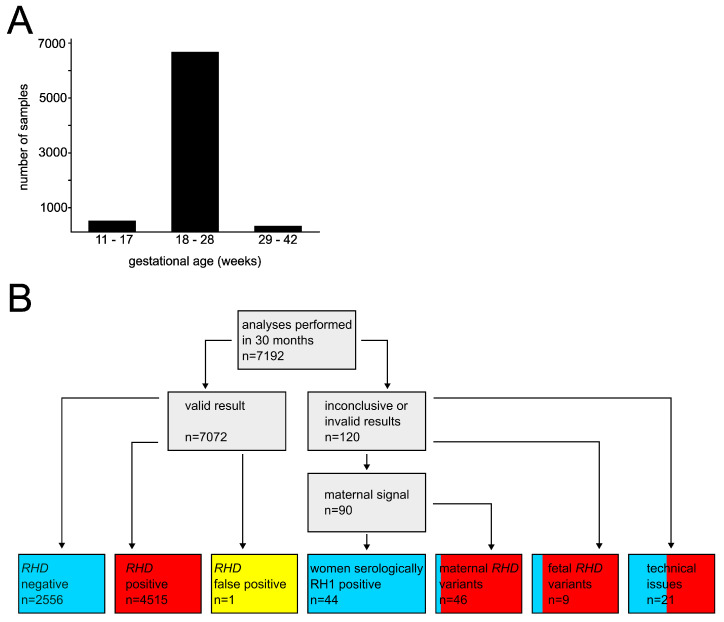
Summary of the results obtained from 7192 analyses between January 2020 and June 2022. (**A**) Distribution of gestational age of analyzed samples. Out of 7192 samples, 421 (5.9%) were collected in gestation weeks 11 to 17, 6523 samples (90.7%) in weeks 18 to 28 and 233 samples (3.2%) in weeks 29 to 42. Gestational age was unknown for 15 samples (0.2%). (**B**) Graphical overview of the results. Blue color is used for fetuses tested as *RHD* negative with no need for RHIG prophylaxis and cases of serologically RH1 positive women not requiring prophylaxis. In contrast, red coloring symbolizes the recommendation for administration of RHIG if a fetus was determined as *RHD* positive or if fetal *RHD* could not be determined due to various reasons. Initially inconclusive or invalid results were further investigated. Dually colored boxes represent the ratio of cases with (red) or without recommendation (blue) for RHIG. One sample was classified false positive, indicated in yellow.

**Table 1 biomedicines-11-02646-t001:** Summary of the results of valid tests.

Total Valid Tests	*RHD* Negative	*RHD* Positive	False Positive
7072	2556	4515	1
	36.1%	63.8%	0.01%

**Table 2 biomedicines-11-02646-t002:** Summary of the results considered inconclusive or invalid due to the detection of maternal *RHD* variants.

Maternal *RHD* Allele/Variant	*RHD* Exons Detected		Number of Individuals	RHIG
	Maternal	Fetal		
*RHD*01W.01*	5, 7		3	no
*RHD*11*	5, 7		2	yes
*RHD*11 + RHD*08N.01*	5, 7		1	yes
*RHD*36*	5, 7		1	yes
*RHD*01W.31*	5, 7		1	yes
*RHD*08N.01*	7	5	1	yes
*RHD*05.07*	7		1	yes
*RHD*960A* ^1^	5, 7		1	yes
ND	5, 7		20	yes
ND	7		5	yes
ND	7	5	10	yes

Maternal *RHD* variants were detected in 46 samples. Further investigation for the identification of variant alleles was carried out in 11 cases, in addition to 35 cases where maternal *RHD* exons were detected but the *RHD* variant was not determined (ND). The amplified maternal and fetal exons are shown. Recommendations for prophylaxis by administration of RH immunoglobulin (RHIG) were given depending on the alleles identified. ^1^ This variant leads to a silent mutation in exon 7 and affects *RHD* expression due to aberrant splicing of the transcript [28].

## Data Availability

Not applicable.

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
