# Peer review of "Fetal RHD Screening in RH1 Negative Pregnant Women: Experience in Switzerland"

_biomedicines, 2023, doi:10.3390/biomedicines11102646_

Round 1

Reviewer 1 Report

The authors described the data of the fetal RHD screening from cell-free DNA in blood collected from pregnant women in the Switzerland population. In this article, they showed that cfDNA screening tests are useful to avoid unnecessary administration of RHIG prophylaxis.

However, in "Materials and Methods" part, the sampling process is not clear. It is written in the article title, but the blood samples were presumably collected from pregnant women who have been confirmed serologically RHD negative. Otherwise, these tests would have no clinical advantage. But those were not written in the sampling process.  In addition, if authors could add more. Similar projects have already started in other countries, so it would be more helpful to the reader if authors showed not only the testing centre and transport or processing delay but also how they received the patient's consent or how testing costs were paid.

English language appropriate and understandable.

Reviewer 2 Report

 Bernd Schimanski and colleagues reported on fatal genotyping for RhD antenatal prophylaxis. 

This is a hot topic and it is very actual. This paper is very well written and very much appreciated for its input and obviously practical implications. This new approach should be worldwide implemented

I believe this paper should be published ASAP in its current format.

Reviewer 3 Report

Thank you for providing the opportunity to review the paper titled "Fetal RHD screening in RH1 negative pregnant women: Experience in Switzerland" In this paper, the authors conducted non-invasive fetal RHD screening through cell-free DNA analysis in a total of 7192 pregnant women who tested RHD-negative. I would like to express the following concerns regarding this paper.

This study comprised two parts. First, the validation study, which involved 208 RHD-negative samples and 307 RHD-positive samples, was described in the Results section. Second, a total of 7192 samples were collected from pregnant women who tested RHD-negative over a 30-month period. The authors should provide a description of the study design above in the Methods section.
